# Controllable hierarchical self-assembly of porphyrin-derived supra-amphiphiles

Shu-Ping Wang[1], Wei Lin[1], Xiaolin Wang[2], Tian-Yong Cen[1], Hujun Xie[3], Jianying Huang[3], Ben-Yue Zhu[1], Zibin Zhang[1], Aixin Song[2], Jingcheng Hao[2], Jing Wu[1] & Shijun Li[1]

Control of self-assembly is significant to the preparation of supramolecular materials and illustration of diversities in either natural or artificial systems. Supra-amphiphiles have remarkable advantages in the construction of nanostructures but control of shape and size of supramolecular nanostructures is still a great challenge. Here, we fabricate a series of supra-amphiphiles by utilizing the recognition motifs based on a heteroditopic porphyrin amphiphile and its zinc complex. These porphyrin amphiphiles can bind with a few guests including Cl⁻, coronene, $C_{60}$, 4,4'-bipyridine and 2,4,6-tri(pyridin-4-yl)-1,3,5-triazine, which are further applied to facilitate the controllable self-assembly. Addition of these guests result in the formation of various supra-amphiphiles with well-defined structures, thus induce the generation of different aggregates. A diverse of aggregation morphologies including nanospheres, nanorods, films, spheric micelles, vesicles and macrowires are constructed upon the influence of specific complexation, which highlights the present work with abundant control on the shapes and dimensions of self-assemblies.

---

[1] College of Material, Chemistry and Chemical Engineering, Hangzhou Normal University, Hangzhou 311121 Zhejiang, China. [2] Key Laboratory of Colloid and Interface Chemistry & Key Laboratory of Special Aggregated Materials, Shandong University, Ministry of Education, Jinan 250100 Shandong, China. [3] Department of Applied Chemistry, Zhejiang Gongshang University, Hangzhou 310018 Zhejiang, China. Correspondence and requests for materials should be addressed to J.H. (email: jhao@sdu.edu.cn) or to S.L. (email: l_shijun@hznu.edu.cn)

Controlling formation of the self-assemblies with ordered and diversified morphologies are important for imitation of natural systems and application of artificial self-assemblies[1–3]. The supra-amphiphiles, a novel research field that combining supramolecular chemistry and traditional amphiphiles, have attracted increasing attention as they can spontaneously assemble to construct a variety of well-defined nanostructures such as micelles, vesicles, tubes, nanorods and nanosheets, which show great application potential in nanodevice, memory storage, drug or gene delivery, protein probe, tissue regeneration, and ion channel regulation[4–12]. The supra-amphiphiles are a class of molecules bearing both hydrophilic and hydrophobic segments that are conjoined by noncovalent bonds or dynamic covalent bonds[13–20] including $\pi-\pi$ stacking[21–24], electrostatic interaction[25,26], hydrogen bonding[27,28], coordination[29], charge-transfer interaction[30], and imine formation[31]. Owing to the dynamic and reversible nature of these covalent or noncovalent interactions, the self-assemblies derived from supra-amphiphiles can be readily controlled by external stimuli including temperature, concentration, solvent, pH, enzyme, redox, ionic strength, and light[4–12,32]. Although the previous studies exhibited the advantages of using dynamic covalent bonds or supramolecular interactions to construct amphiphiles, fine control in the shape, dimension, and size of supramolecular nanostructures is still a great challenge.

Due to their distinctive $\pi$-conjugated architectures, excellent photosensitivity and biochemical functionality, porphyrins have been widely used as building blocks in supramolecular self-assemblies[33–41]. Functional groups can be easily introduced into the porphyrin moieties, endowing the obtained supramolecular systems with interesting properties. $\alpha,\alpha,\alpha,\alpha$-*meso*-Tetrakis(2-aminophenyl)porphyrin ($\alpha,\alpha,\alpha,\alpha$-H$_2$TamPP)[42,43] and its derivatives possess unique "picket fence" topological structures and

attractive features. They have been previously used as binucleating ligands[44–47], building blocks for cages[48], anion receptors[49–51], synthetic models for oxygen binding hemoproteins[52] and cytochrome oxidase[53,54], as well as chloroperoxidase analogs[55]. The unique structure of $\alpha,\alpha,\alpha,\alpha$-H$_2$TamPP makes it very suitable as a building block to prepare supra-amphiphiles for the controllable self-assembly. Herein, we design and synthesize a porphyrin-based heteroditopic amphiphile by utilizing $\alpha,\alpha,\alpha,\alpha$-H$_2$TamPP. This unilateral porphyrin amphiphile and its zinc complex consist of a hydrophobic urea-derived porphyrin and four hydrophilic oligoether chains that point to the same side of the porphyrin plane, which can not only be served as receptors to bind with inorganic or organic guests including Cl$^-$, coronene, C$_{60}$, and pyridyl ligands, but also be applied in the construction of supra-amphiphiles to realize the fine control of amphiphilic self-assembly upon the addition of different guests. Addition of these specific guests result in the fabrication of supra-amphiphiles with different well-defined topological structures, thus induce the change of self-assembly morphologies.

## Results

**Synthesis and characterization.** As illustrated in Fig. 1, the unilateral porphyrin amphiphiles were synthesized in several steps. The compound $\alpha,\alpha,\alpha,\alpha$-H$_2$TamPP **1** could be easily prepared by cyclization from 2-nitrobenzaldehyde and pyrrole, reduction, and then atropisomerization[56], according to the procedures described in the Methods section. The four amino groups of **1** were transformed to isocyanates with triphosgene to generate the compound **2**, while reaction of *p*-acetaminophenol with tetraethylene glycol monotosylate **3** produced tetraethylene glycol-substituted aniline **4**. The isocyanate **2** was then reacted with excess **4** in dichloromethane to furnish the amphiphilic porphyrin

**Fig. 1** Synthesis of amphiphilic porphyrin **5** and its Zn-complex **6**

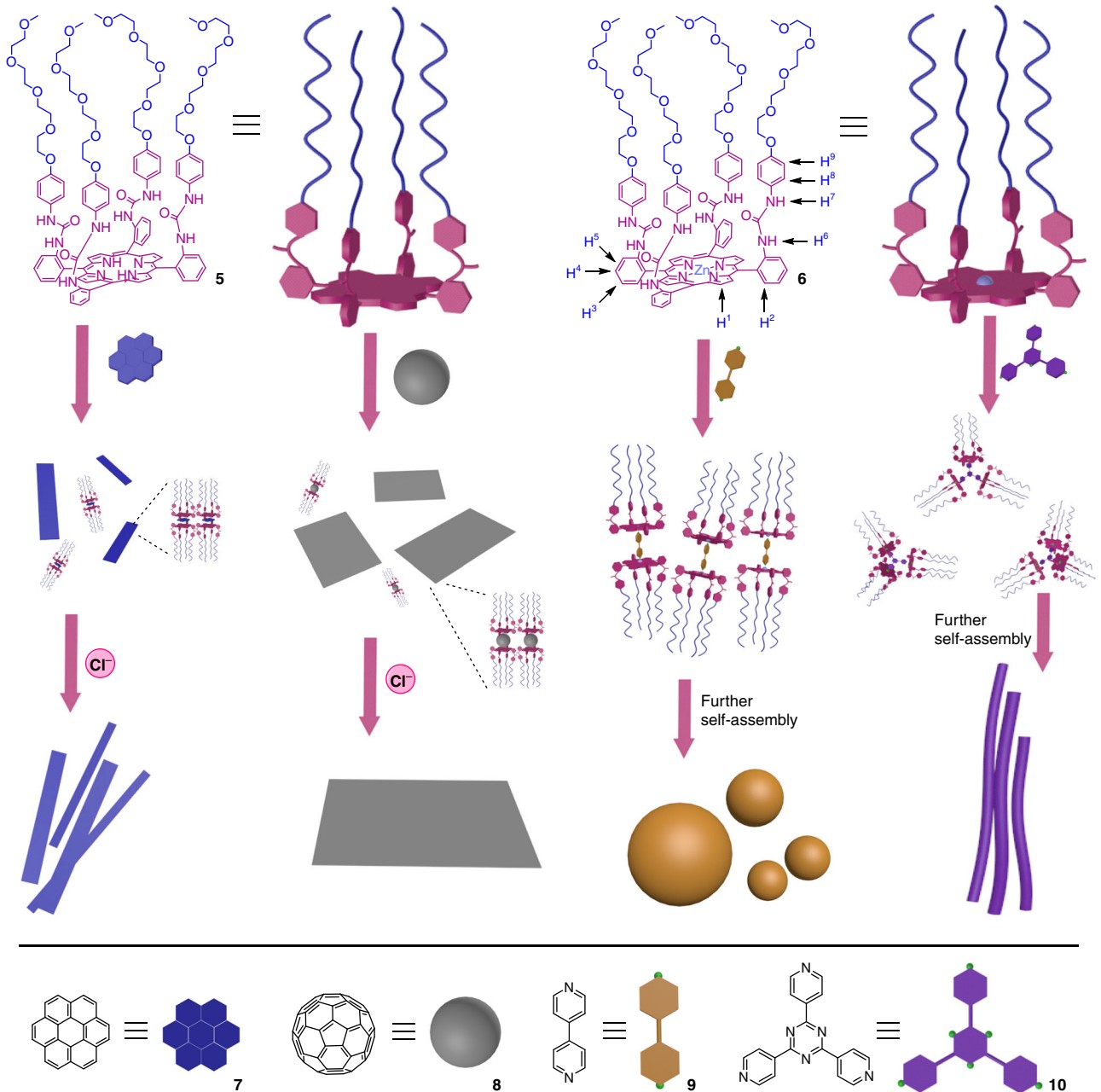

**Fig. 2** Hierarchical self-assembly of the amphiphilic porphyrin **5** and Zn-porphyrin **6** controlled by the addition of different guests

**5** in 68% yield. Complexation of **5** with Zn(OAc)$_2$·2H$_2$O in chloroform afforded the Zn-porphyrin derivative **6** in 73% yield. Both **5** and **6** were fully characterized by $^1$H NMR, $^{13}$C NMR, high-resolution mass spectroscopy (HRMS), infrared spectroscopy (IR), UV-vis spectroscopy, and fluorescence spectroscopy (Supplementary Figs. 1–8 and Supplementary Figs. 17–20). Based on the UV-vis transmittance, the critical aggregation concentrations (CACs) of the amphiphiles **5** and **6** in aqueous solutions were measured to be ~$7.0 \times 10^{-6}$ M and $9.0 \times 10^{-6}$ M, respectively (Supplementary Figs. 23, 24).

In order to visualize the spacial orientation of the amphiphilic porphyrin, molecular simulation of **5** was carried out. As shown in the calculated structure of **5** (Supplementary Fig. 9), the four long oligoether chains are oriented toward the same side of the porphyrin plane, which can act as the hydrophilic part to avoid exposing hydrophobic areas within a hydrophilic environment. The oligoether chains enforce the amphiphilic porphyrin

molecules to be arranged together to form dimeric structures with the hydrophobic porphyrin planes face-to-face and the hydrophilic sides up or down, which create the possibility of controlling aggregation morphology through specific complexation interactions (Fig. 2).

**Self-assembly of 5 and influence of Cl$^-$ and/or coronene**. With the amphiphilic porphyrin derivatives in hand, the self-assembly of **5** in tetrahydrofuran (THF) was firstly investigated by TEM observations. As self-assemblies of most organic molecules usually behave in homogeneous solution, the heteroditopic amphiphile **5** aggregated into small spheres of several nanometers in diameter in THF (Fig. 3a). When 4.0 equivalents of Cl$^-$ with the counterion of $n$-Bu$_4$N$^+$ was added, the **5** solution in THF self-assembled into much larger spherical particles of 70–150 nm in diameter (Fig. 3b), which was also supported by dynamic light scattering (DLS) data (Fig. 3c). The use of bulky $n$-Bu$_4$N$^+$ cation

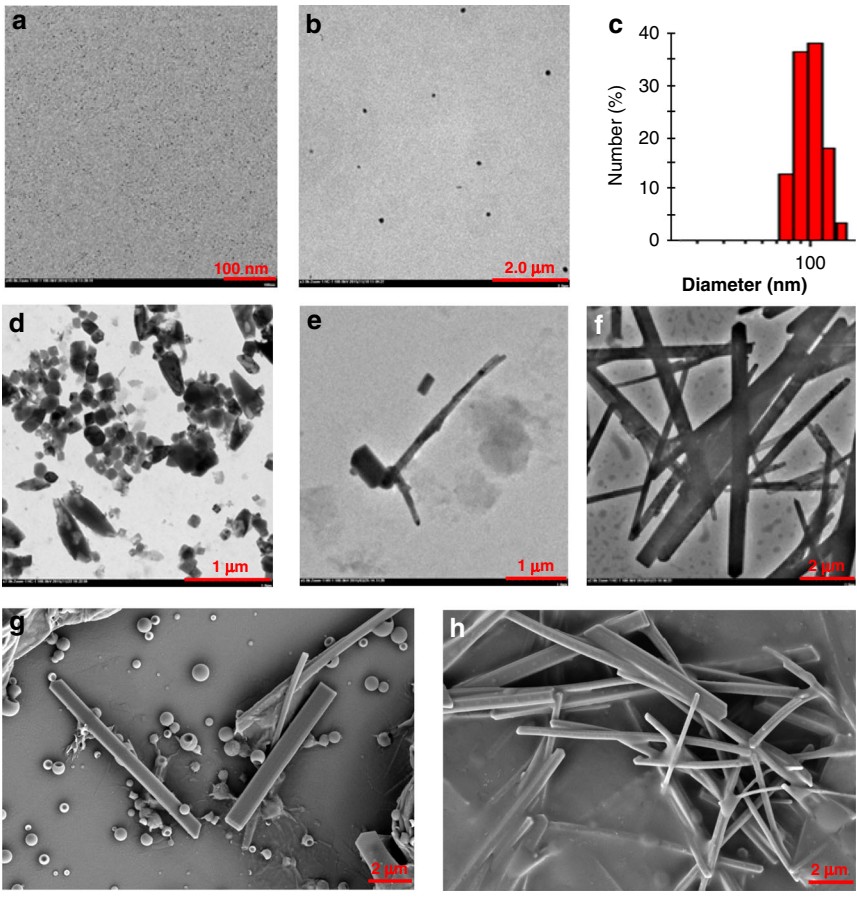

**Fig. 3** Electron microscope images of self-assemblies formed from **5** or **5₂•7**, and DLS data of **5**. **a** TEM image of **5** (1.00 × 10⁻⁴ M). **b** TEM image of **5** (1.00 × 10⁻⁴ M) with the addition of 4 equiv $n$-Bu₄NCl in THF. **c** DLS data of **5** (1.00 × 10⁻⁴ M) with the addition of 4 equiv $n$-Bu₄NCl in THF. **d** TEM image of **5** (1.00 × 10⁻⁴ M) with the addition of 4 equiv $n$-Bu₄NCl in aqueous solution. **e** TEM image of **5₂•7** (1.00 × 10⁻⁴ M) in aqueous solution. **f** TEM image of **5₂•7** (1.00 × 10⁻³ M) with the addition of 4 equiv $n$-Bu₄NCl in aqueous solution. **g** SEM image of **5₂•7** (1.00 × 10⁻³ M) in aqueous solution. **h** SEM image of **5₂•7** with the addition of 4 equiv $n$-Bu₄NCl (1.00 × 10⁻³ M) in aqueous solution

can benefit complexation of Cl⁻ with the urea groups on **5** attributing to its good solubility in organic solvents and its separated ion pair characteristics[57,58].

The self-assembly behavior of the amphiphilic porphyrin in aqueous solution was subsequently investigated. The samples in water were prepared by adding a solution of the amphiphile or supra-amphiphile in an organic solvent into water and then volatilizing the organic solvent. THF was found the best choice for self-assembly among the screened organic solvents including acetone, acetonitrile, chloroform, dichloromethane, and ethanol, probably attributing to the good solubility of amphiphilic porphyrin in THF as well as good miscibility of THF with water. As shown in the TEM images (Supplementary Fig. 33), the amphiphile **5** self-assembled into irregular supramolecular polymers, with the assistance of the bulky hydrophobic head and four flexible oligoether chains, as well as intermolecular hydrogen bonds between the urea groups. When 4.0 equivalents of $n$-Bu₄NCl that can be complexed by urea units of **5** were added into the aqueous solution, different assembly morphology of **5** was observed in the TEM images. As shown in Fig. 3d, large pellets with random shapes and sizes formed. Although the shape is not uniform, it is more regular than the assembly morphology of **5** itself (Fig. 3d vs Supplementary Fig. 33). This can be attributed to the strong affinity of ureas on the amphiphilic porphyrin toward the spherical Cl⁻ anions via complexation that increases intermolecular interactions of **5** to promote the formation of larger self-assemblies. The complexation between

the amphiphiles and Cl⁻ was confirmed by the NMR titration studies. Obvious change of chemical shifts in the ¹H NMR spectra was observed after the addition of $n$-Bu₄NCl to the solution of either **5** or **6**, while the tight ion pair NaCl could not complex with the amphiphiles (Supplementary Figs. 10, 11).

Furthermore, it was found that the size and shape of aggregation morphology of the heteroditopic amphiphile **5** could be tuned by binding aromatic molecules with different geometries. When 0.5 equivalents of coronene **7** was added to a solution of **5**, the flat coronene inserted into the middle of two porphyrin molecules to form a sandwich-like complex. Their complexation was attested by the change of chemical shifts in the ¹H NMR spectra (Fig. 4) and existence of nuclear Overhauser effect (NOE) between the protons on **5** and the proton on **7** in NOESY NMR spectrum (Supplementary Fig. 14). The change of UV-vis and fluorescence spectroscopy (Supplementary Figs. 17, 19) provided further evidence for the complexation. Obvious fluorescence quench was observed upon the addition of 0.5 equivalents of **7** (Supplementary Fig. 19). A 2:1 binding stoichiometry for the complexation between **5** and **7** was determined by a Job plot based on the UV-vis titration experiments (Supplementary Fig. 21). It can be concluded that the four long oligoether chains grafted on the porphyrin segment enforce the hydrophobic porphyrin plane to face toward coronene to form a sandwich complex **5₂•7**. The binding constants of **5₂•7** in THF were further measured on the basis of fluorescence titration data. The linearity of the Scatchard plot (Supplementary Fig. 29) indicates that the

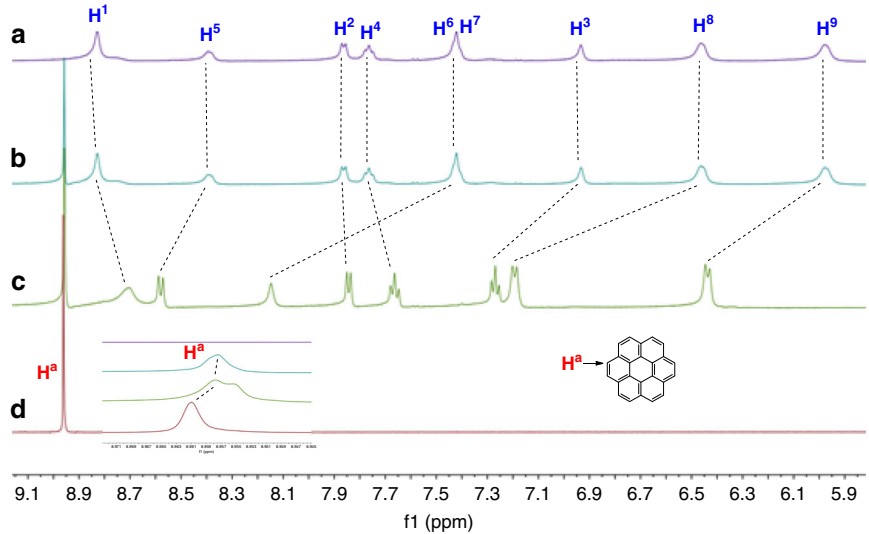

**Fig. 4** Partial $^1$H NMR (500 MHz, THF-$d_8$, 25 °C) spectra of **5**, **5$_2$·7**, and **7**. **a** $^1$H NMR spectrum of **5**. **b** $^1$H NMR spectrum of **5$_2$·7**. **c** $^1$H NMR spectrum of a mixture of **5$_2$·7** and $n$-Bu$_4$NCl. **d** $^1$H NMR spectrum of **7**

complexation between **5** and **7** is statistical, that means, the two complexation sites behave independently[59,60]. Furthermore, the slope of the Scatchard plot is proportional to the negative of the average association constant ($K_{av} = (K_1 + K_2)/2$), and the intercept is $K_{av}$. Therefore, $K_{av}$ was determined to be 5.3 ($\pm$0.1) $\times$ 10$^4$ M$^{-1}$ for **5$_2$·7**. Since $K_1/K_2 = 4:1$ for statistical systems ($K_1 = [5\cdot7]/\{[5][7]\}$ and $K_2 = [5_2\cdot7]/\{[5\cdot7][5]\}$), $K_1$ and $K_2$ were calculated to be 8.5 ($\pm$0.2) $\times$ 10$^4$ M$^{-1}$ and 2.1 ($\pm$0.1) $\times$ 10$^4$ M$^{-1}$, respectively. The high binding ability ensures the supra-amphiphile stable during the self-assembly. In spite of the fact that uncomplexed species usually increase at low concentrations, hydrophobic interactions existed during self-assembly in aqueous solutions would greatly enhance the stability of the supra-amphiphiles.

Although the aggregation morphologies of the mixture of **5** and **7** are relatively disorganized with the existence of a few nanorods (Fig. 3e, g, and Supplementary Figs. 34, 35a–d), well-organized morphologies of rectangular nanorods were found when 4.0 equivalents of $n$-Bu$_4$NCl was further added to the mixture of **5** and **7** (Fig. 3f, h, and Supplementary Fig. 35e–h and 36). This morphological transformation is proposed due to the strong complexation between **5$_2$·7** and Cl$^-$, which was proved by the significant change of chemical shifts of the protons on **5** after the addition of Cl$^-$ into a solution of **5$_2$·7** (Fig. 4). The Cl$^-$ anions bound by the amphiphilic porphyrin enhanced intermolecular interactions between the neighboring **5$_2$·7** complexes, making the whole supramolecular assemblies more orderly. This phenomenon provides another direct evidence that the addition of Cl$^-$ can promote the formation of well-arranged supramolecular structures. The concentration of the solution has a little influence on the size of aggregates. In the presence of Cl$^-$, the supramolecular nanorods self-assembled at a concentration of 1 mM have a range of length from 5 to 10 µm with width between 300 and 600 nm (Fig. 3f, h), while the nanorods formed at a concentration of 0.1 mM are relative smaller with a range of length from 1 to 3 µm and width from 50 to 120 nm (Supplementary Fig. 35g, h).

**Influence of C$_{60}$ and Cl$^-$ on the self-assembly of 5.** It was further demonstrated that addition of another aromatic guest C$_{60}$ **8** with a different geometry could also induce a dramatic change in the aggregation morphology of the amphiphilic porphyrin **5**. Similar

to coronene, one molecule of C$_{60}$ can be bound by two molecules of **5** to form a sandwich-like complex. The complexation between **5** and **8** was ascertained by the obvious shift of NMR signals of protons on **5** (Fig. 5), as well as the change of UV-vis and fluorescence spectroscopy (Supplementary Figs. 17, 19). A 2:1 binding stoichiometry was calculated by a Job plot from the UV titration data (Supplementary Fig. 22). On the basis of fluorescence titration experiments, the two binding sites were found to be independent or statistical, and the binding constants $K_1$ and $K_2$ of **5$_2$·8** in toluene were calculated to be 3.2 ($\pm$0.2) $\times$ 10$^3$ M$^{-1}$ and 8.0 ($\pm$0.4) $\times$ 10$^2$ M$^{-1}$, respectively (Supplementary Fig. 30). As shown in the TEM (Fig. 6a and Supplementary Fig. 37) and SEM images (Fig. 6e and Supplementary Fig. 38a, b), adding 0.5 equivalents of **8** to a solution of **5** resulted in the generation of lamellar aggregates that were totally different from the assembly morphologies of both **5** and **5$_2$·7**. The formed lamellar structures were up to square micrometers in area and the surface of the lamellae seemed to be obscure looking like half-baked membrane accompanied with a few spheric aggregates. It can be deduced that the oligoether segments as the hydrophilic parts presented on the outer surface of the aggregates, whereas **8** and the two hydrophobic porphyrin segments gathered together like a sandwich biscuit in the core. The thickness of the monolayer lamellae was measured ~10 nm by atomic force microscopy (AFM; Supplementary Fig. 39). The self-assembly of **5$_2$·8** could also be affected by the complexation of Cl$^-$. The obvious change of chemical shifts of protons on **5** upon the addition of Cl$^-$ testified the existence of complexation between **5$_2$·8** and Cl$^-$ (Fig. 5). As the TEM (Fig. 6b and Supplementary Fig. 40), SEM (Fig. 6f and Supplementary Fig. 38c, d) and AFM (Fig. 6c, d) images revealed, the morphologies became large, smooth, and solid films with sleek edges, indicating that Cl$^-$ can promote aggregation of **5$_2$·8**. The curls and wrinkles on the films might be a consequence of the drying process during sample preparation. The driven force of the formation of films could be attributed to the hydrophobic interaction, $\pi-\pi$ stacking among the neighboring C$_{60}$ molecules[67,68], and intermolecular hydrogen bonds among the urea groups or the bridged complexation of urea groups with Cl$^-$ (Fig. 6g). Herein, $\pi-\pi$ stacking among the C$_{60}$ molecules plays a significant role for the formation of films. This strategy provides a highly efficient way to fabricate monolayers of fullerenes, which is quite attractive for the application in photovoltaic materials.

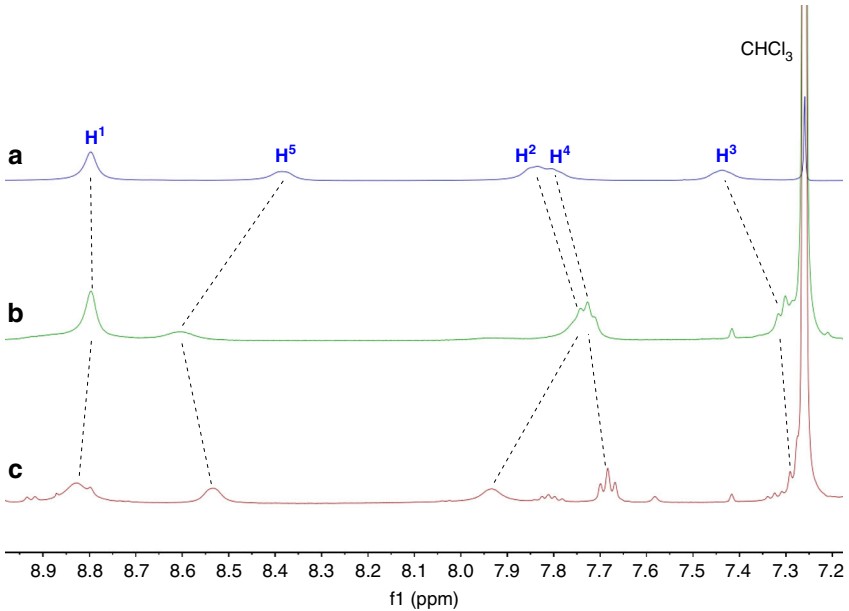

**Fig. 5** Partial $^1$H NMR (500 MHz, CDCl$_3$, 25 °C) spectra of **5** and **5$_2$•8**. **a** $^1$H NMR spectrum of **5**. **b** $^1$H NMR spectrum of **5$_2$•8**. **c** $^1$H NMR spectrum of a mixture of **5$_2$•8** and *n*-Bu$_4$NCl

**Self-assembly of 6 and influence of the pyridyl guests**. Pyridyl compounds are good alternatives to be used as the guests that can easily coordinate with metalloporphyrins[35,39,40]. The fabrication of supra-amphiphiles upon the addition of different pyridyl guests was investigated by using Zn-porphyrin **6** as the amphiphile. When 0.5 mol equivalents of 4,4′-bipyridine **9**, or 1/3 mol equivalents of 2,4,6-tri(pyridin-4-yl)-1,3,5-triazine (TPT) **10** was added to a **6** solution, the central zinc ion coordinated with the N atoms on pyridyl guests to generate the supra-amphiphilic complexes **6$_2$•9** and **6$_3$•10**. The coordination between **6** and the two pyridyl guests was confirmed by $^1$H NMR, 2D NOESY NMR, UV-vis, and fluorescence spectroscopy. Change of the chemical shifts of protons on both **6** and the pyridyl compounds (Supplementary Figs. 12, 13) as well as the NOEs between the protons on **6** and the guests (Supplementary Figs. 15, 16) were found. Absorption of the sharp peak at 428 nm that is assigned to the Soret band of the porphyrin moiety reduced significantly after **6** was coordinated by the pyridyl compounds, while the peak at 556 nm increased (Supplementary Fig. 18). Meanwhile, an increase in the fluorescence intensity at 654 nm and a slight red shift of this emission peak were observed after the addition of **9** or **10** (Supplementary Fig. 20). Based on fluorescence titration studies, the Scatchard plots were resulted and independent binding behavior for both **6$_2$•9** and **6$_3$•10** was demonstrated (Supplementary Figs. 31, 32). The association constants $K_1$ and $K_2$ of **6$_2$•9** in THF were measured to be 1.6 ($\pm$0.1) $\times$ 10$^3$ M$^{-1}$ and 4.0 ($\pm$0.2) $\times$ 10$^2$ M$^{-1}$, respectively, and $K_1$, $K_2$, and $K_3$, of **6$_3$•10** in THF were 2.4 ($\pm$0.1) $\times$ 10$^3$ M$^{-1}$, 8.1 ($\pm$0.1) $\times$ 10$^2$ M$^{-1}$ and 2.7 ($\pm$0.1) $\times$ 10$^2$ M$^{-1}$, respectively. These considerable binding affinities are conducive to the formation of stable supra-amphiphiles.

When **9** was used as the guest, a dumbbell-shaped supra-amphiphile **6$_2$•9** was formed, which is composed of a stiff hydrophobic segment including two coordinated zinc-porphyrins in the core and two parts of flexible hydrophilic oligoether segments at the two sides that are exposed to the aqueous environment. TEM images showed that the supramolecular amphiphile **6$_2$•9** self-assembled into stable micelles with different sizes. As shown in Fig. 7a, b and Supplementary Fig. 41, immense spheric micelles with the diameter of 0.3–0.8 μm were obtained

from a 1 mM aqueous solution of **6$_2$•9** cast onto micrograph TEM grids. The observed preservation of immense spheric micelles suggests that the amphiphilic complexes stacked together with the aid of hydrogen bonding, π−π stacking, and hydrophobic interactions to form spheres (Fig. 7e). After *n*-Bu$_4$NCl was added into a solution of **6$_2$•9**, the aggregation morphology changed from the neat spheres to relatively irregular vesicles at the same sample-preparation conditions (Fig. 7c, d and Supplementary Fig. 42). This change of morphology is probably caused by the enhancement of intermolecular interactions among the neighboring **6$_2$•9** complexes through chloride anions, resulting in that the spheric self-assemblies were compressed to form irregular vesicles.

As different with the above dumbbell-shaped amphiphile, the trigonal amphiphilic complex **6$_3$•10** was constructed by one TPT and three Zn-porphyrins in the hydrophobic center surrounded by the flexible hydrophilic oligoethers. Completely different aggregation behavior was observed when **6$_3$•10** was self-assembled in aqueous solution. The TEM images of **6$_3$•10** clearly show microwire or microfiber aggregates with the diameter of ~300–500 nm (Fig. 8a, b and Supplementary Fig. 43). As shown in the SEM images, an overview of the whole microwires with the length of 300–500 μm was observed (Supplementary Fig. 44). The TEM and SEM images revealed that microwires with about the same sizes and morphologies were observed (Fig. 8c and Supplementary Figs. 45, 46) when 4.0 equivalents of *n*-Bu$_4$NCl was added to a solution of **6$_3$•10**, indicative of that Cl$^-$ almost has no effect on the aggregation behavior of the amphiphilic complex **6$_3$•10**. This is probably because the intermolecular interactions among the **6$_3$•10** complexes are strong enough to maintain the stable microwires via multiple hydrogen bonds between six urea groups on each side of **6$_3$•10**, and the addition of Cl$^-$ could not apparently alter the arrangement of **6$_3$•10** during the self-assembly.

**Investigation on self-assembly mechanism**. In order to discern whether the above aggregates are thermodynamically stable, self-assemblies over longer time and at higher temperature were carried out. In all cases of **5$_2$•7**, **5$_2$•8**, **6$_2$•9**, and **6$_3$•10**, TEM

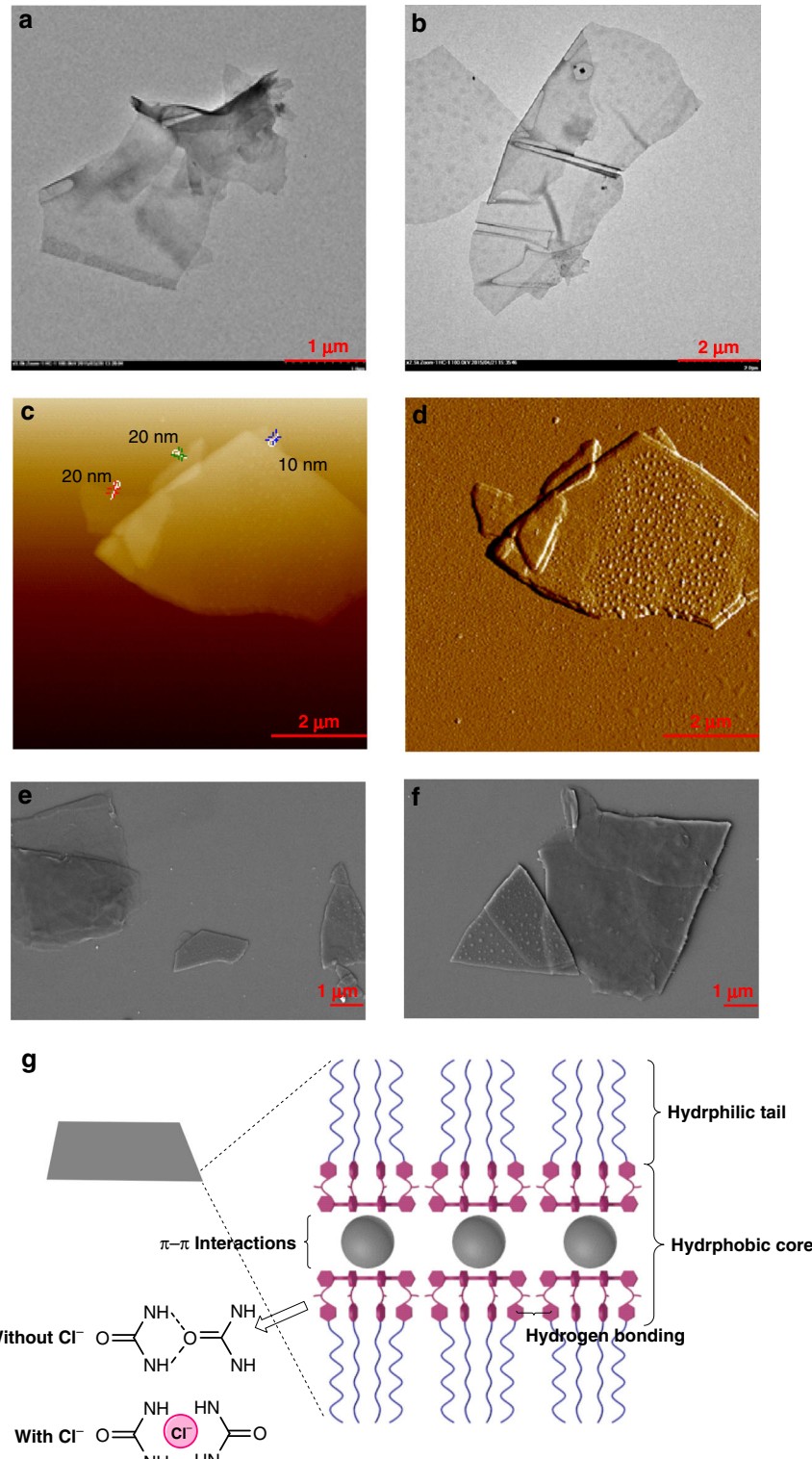

**Fig. 6** Electron microscope images of self-assemblies formed from **5₂•8**. **a** TEM image of **5₂•8** ($1.00 \times 10^{-4}$ M) in aqueous solution. **b** TEM image of **5₂•8** ($1.00 \times 10^{-4}$ M) with the addition of 4 equiv *n*-Bu₄NCl in aqueous solution. **c, d** AFM images of **5₂•8** ($1.00 \times 10^{-3}$ M) with the addition of 4 equiv *n*-Bu₄NCl in aqueous solution. **e** SEM image of **5₂•8** ($1.00 \times 10^{-4}$ M) in aqueous solution. **f** SEM image of **5₂•8** ($1.00 \times 10^{-4}$ M) with the addition of 4 equiv *n*-Bu₄NCl in aqueous solution. **g** Schematic representation of the self-assembly of **5₂•8**

images of the samples prepared after 3 and 7 days at room temperature, as well as the samples prepared at 60 °C showed no significant difference in aggregation morphologies (Supplementary Figs. 47–50), which implies the observed aggregates are thermodynamically stable instead of kinetically trapped

structures. Based on the UV-vis transmittance, the CACs of supra-amphiphiles **5₂•7**, **5₂•8**, **6₂•9**, and **6₃•10** in aqueous solutions were measured to be $5.5 \times 10^{-6}$ M, $8.0 \times 10^{-6}$ M, $1.8 \times 10^{-5}$ M, and $1.7 \times 10^{-5}$ M, respectively (Supplementary Figs. 25–28). These supra-amphiphiles with different geometry have rather

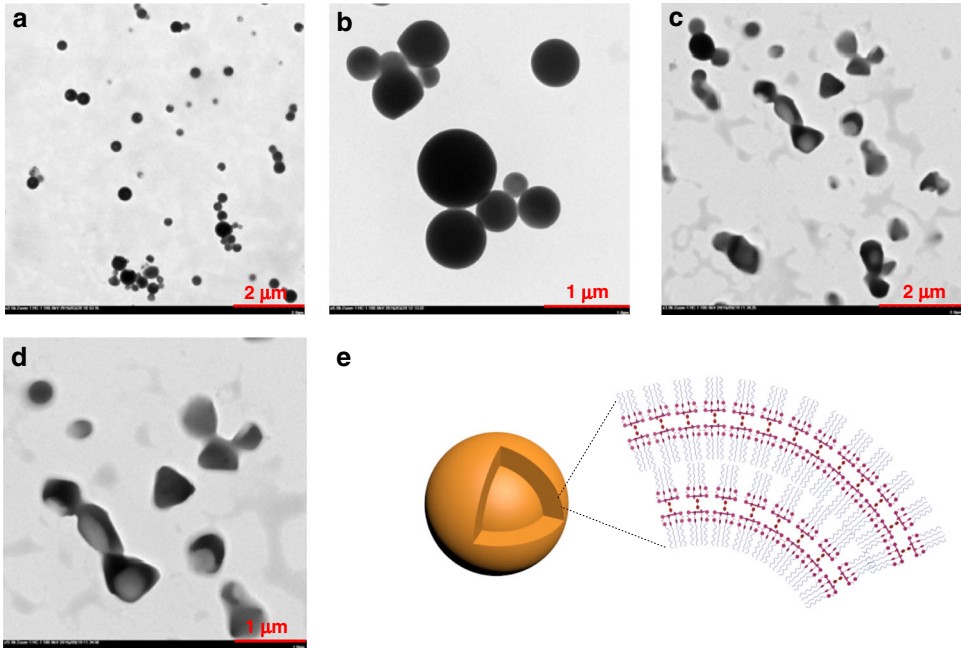

**Fig. 7** TEM images of self-assemblies formed from **6₂•9**. **a, b** TEM images of **6₂•9** (1.00 × 10⁻³ M) in aqueous solution. **c, d** TEM images of **6₂•9** (1.00 × 10⁻³ M) with the addition of 4 equiv n-Bu₄NCl in aqueous solution. **e** Schematic representation of the self-assembly of **6₂•9**

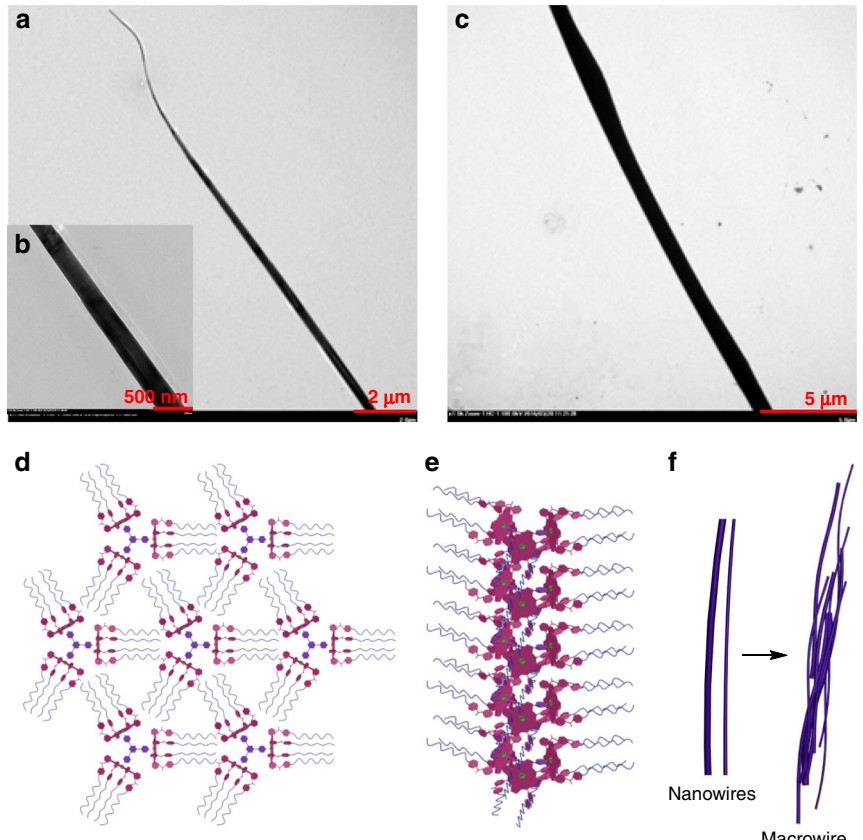

**Fig. 8** TEM images of self-assemblies formed from **6₃•10**. **a, b** TEM images of **6₃•10** (1.00 × 10⁻³ M) in aqueous solution. **c** TEM image of **6₃•10** (1.00 × 10⁻³ M) with the addition of 4 equiv n-Bu₄NCl in aqueous solution. **d** Schematic representation of the cross section of the nanowires. **e** Stacking mode in the nanowires. **f** Schematic representation of macrowires consisting of nanowires

close CAC values, which indicates that the structural characteristics and supramolecular interactions, but not the difference in hydrophilicity or lipophilicity, play important roles on the formation of varied aggregation morphologies.

To further understand the mechanism of microwire formation, TEM observations were performed at different stages of self-assembly (Fig. 9). A solution of the amphiphilic complex **6₃•10** at 2 mM concentration in THF was firstly prepared.

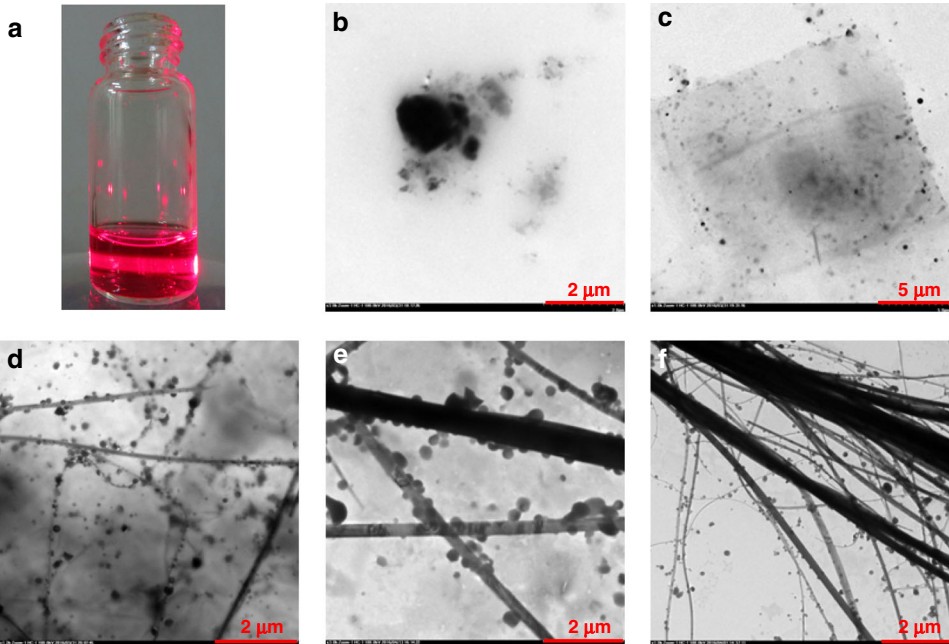

**Fig. 9** Structural changes during the self-assembly of $6_3 \cdot 10$ represented with TEM images investigated at different stages. **a** The Tyndall effect observed immediately when water was added to a $6_3 \cdot 10$ solution in THF. **b** At initial stage, nanoaggregates formed predominantly after 1 h. **c** The stage that merged into membrane with the volatile of THF after 3 h. **d** Nanowires around with some nanoparticles appeared after 7 h. **e** Microwires formed by conglutinating the nanoparticles to the nanowires to increase their diameters after 14 h. **f** Eventually, the nanowires and microwires bonded together to form stable wire bundles after 3 days

Tyndall effect (Fig. 9a) was captured after an equal volume of water was added, indicating the formation of colloids or nanoaggregates[61]. At the initial stage, formation of irregular micelles was observed in the TEM images (Fig. 9b). As time goes on, THF was gradually volatilized, resulting in the formation of membranes (Fig. 9c). With further increase of time, the appearance of nanowires in the membranes with many nanoparticles adhered to the wires was found (Fig. 9d). It was speculated that the hydrophobic interaction among the trigonal coordinated Zn-porphyrin cores enhanced with the volatilization of THF. The nanowires were further assembled to form macrowires or wire bundles with larger diameters (Fig. 9e, f), which may comprise numerous nanowires to reduce the hydrophobic interaction and increase the orderliness of the entire system. It is interesting that the nanoparticles aggregating adhesively around the nanowires appear to increase the diameter of wires with the loss of organic solvents. We inferred that the formation of macrowires is a gradually stabilizing process, which is also supported by the SEM micrographs for the self-assembly process studies (Supplementary Fig. 51).

The phenomena described above represent that the self-assembly structure changed successively, from irregular nanoaggregates, membranes, nanowires to regular macrowires. This change process along with the volatilization of THF can be rationalized by considering the strong tendency of $6_3 \cdot 10$ to form anisotropic arrangement and consequent space filling requirements[62] (Fig. 8d–f). It is speculated that the nanowires formation is driven by the collapse of the amphiphilic complex $6_3 \cdot 10$, maximizing interactions between the hydrophilic oligoether chains and causing a release of solvation entropy[63,64]. The primary driving force responsible for the formation of nanowires is proposed to be the energy balance between repulsive interactions among the adjacent hydrophilic oligoether chains and attrahent hydrogen bonds among ureas. The TPT moieties in the center of the discrete aggregates have a strong tendency to

arrange with an anisotropic orientation via $\pi-\pi$ stacking interaction[65]. Owing to an extreme steric problem of the trigonal complex $6_3 \cdot 10$ to be packed radially, the parallel arrangement of TPT is expected to be the best way to reduce the steric repulsion among them. To reduce the unfavorable contact between hydrophobic edges and the aqueous solution, the nanowires bonded together to form more stable macrowires or wire bundles.

An attempt to probe the internal structures of the previous aggregates was made with selected area electron diffraction (SAED) as a tool[66]. As revealed by the HRTEM image and strong diffraction patterns (Fig. 10a, b), the nanorods assembled from $5_2 \cdot 7$ show good crystallinity of the microsized crystals. The membranes constructed from $5_2 \cdot 8$ have weak crystallinity (Fig. 10c, d) while the micelles assemblied from $6_2 \cdot 9$ have no diffraction and thus are not crystalline (Fig. 10e, f). The consecutive diffraction patterns (Fig. 10h) from the microwires that aggregated by $6_3 \cdot 10$ implies the existence of superstructures or subunits that can also be directly observed in the enlarged HRTEM picture (Fig. 10g), the results of which provide further solid evidence for the self-assembly mechanism deduced from the TEM and SEM observations at different stages. Overall, the varied diffraction patterns from the aggregates with different morphologies suggest their different arrangement modes of building molecules.

## Discussion

A amphiphilic porphyrin with four oligoether chains and its corresponding Zn-coordinated derivative were synthesized. The amphiphilic porphyrin can self-assemble into nanostructures, while the self-assembly morphologies changed dramatically upon the addition of coronene, $C_{60}$, and/or $Cl^-$. As the distance between two hydrophobic porphyrin planes enlarged by insertion of the aromatic guests with different sizes and shapes, various aggregation morphologies such as nanospheres, nanorods, and

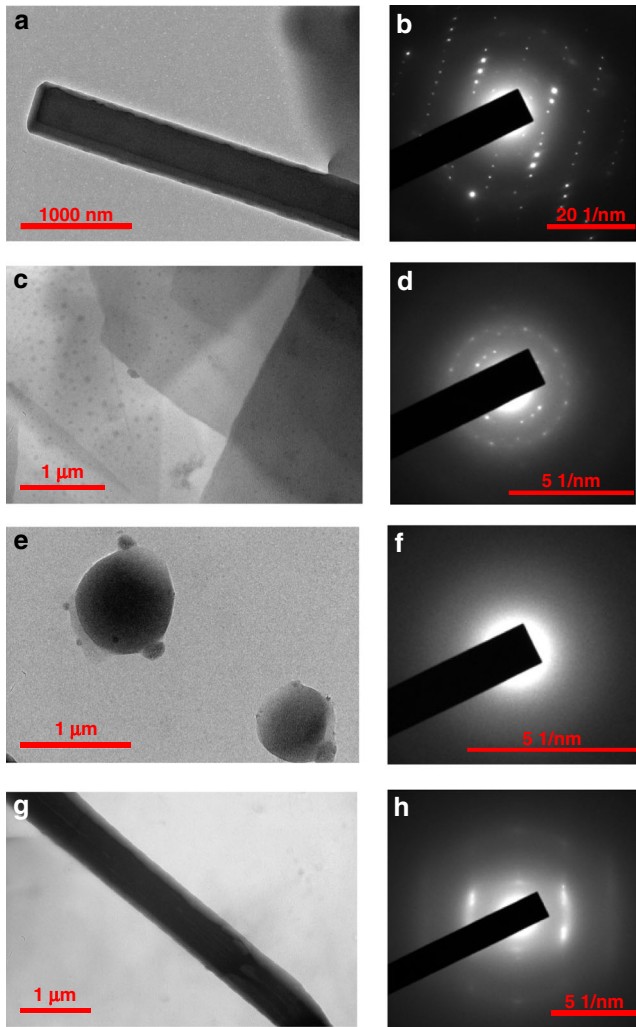

**Fig. 10** HRTEM images and the corresponding SAED patterns. **a** HRTEM image and **b** SAED pattern of **5₂•7** (1.00 × 10⁻³ M) with the addition of 4 equiv *n*-Bu₄NCl. **c** HRTEM image and **d** SAED pattern of **5₂•8** (1.00 × 10⁻³ M) with the addition of 4 equiv *n*-Bu₄NCl. **e** HRTEM image and **f** SAED pattern of **6₂•9** (1.00 × 10⁻³ M). **g** HRTEM image and **h** SAED pattern of **6₃•10** (1.00 × 10⁻³ M) with the addition of 4 equiv *n*-Bu₄NCl

films were formed as a result of the influence of specific complexation interactions. Self-assembly of the amphiphilic Zn-porphyrin derivative could also be controlled to form spheric micelles, vesicles and macrowires after coordination with 4,4′-bipyridine or 2,4,6-tri(pyridin-4-yl)-1,3,5-triazine. The presented studies provide a new strategy to construct supra-amphiphiles by using porphyrin-based recognition motifs, and intensively exhibit the advances of supra-amphiphiles for morphological control with the convenient addition of different guests, which will have great potential applications in smart materials and biological systems.

## Methods

**Materials and general methods**. All chemical reagents were commercially available and used as received. Dichloromethane was dried and redistilled with CaH₂. Other solvents were employed as purchased. NMR spectra were collected on a Bruker AVANCE DMX-500 spectrometer at room temperature. Chemical shifts (δ) are given in ppm and referenced to the standard TMS or residual solvent peaks. Mass spectra were recorded on an Agilent Technologies 6530 Q-TOF mass spectrometer with ESI resource. Dynamic light scattering (DLS) were recorded on a Malvern Nanosizer S instrument at room temperature. Transmission electron microscope (TEM) was performed at 120 kV using JEM-2010 or JEM-1010. Scanning electron microscope (SEM) was done with a JSM-6390LV or TM-1000

using second electron images. High-resolution transmission electron microscope (HRTEM) was carried out on a JEM-2100F at room temperature. Atomic force microscope (AFM) was carried out on a Bruker ICON3-sys. UV-vis spectra were measured with a Hitachi UH-5300 at room temperature. Fluorescence spectra were recorded on a Perkin Elmer LS55 at room temperature. Fourier transform infrared (FTIR) spectroscopy was carried out on a Thermo Scientific Nicolet iS50 at room temperature and only noteworthy absorptions (in cm⁻¹) are listed.

**Synthesis of compound 1**. Pyrrole (12.6 mL, 182 mmol) was added dropwise to a solution of 2-nitrobenzaldehyde (25 g, 165 mmol) in acetic acid (500 mL) under reflux[56]. After the mixture was stirred for 0.5 h, 75 mL of chloroform was added. The mixture was cooled to 0 °C to precipitate the product. The solid was filtered and washed with chloroform, and then dried at 100 °C to furnish *meso*-tetrakis(2-nitrophenyl)porphyrin (3.9 g, 12%). A solution of SnCl₂•2H₂O (10.2 g, 45.0 mmol) in conc. hydrochloric acid (40 mL) was added dropwise to a solution of *meso*-tetrakis(2-nitrophenyl)porphyrin (2.4 g, 3.0 mmol) in conc. hydrochloric acid (100 mL) at room temperature under N₂ atmosphere. The reaction was carried out for 3 h at room temperature, and was then continued at 65 °C for 0.5 h. After cooling to room temperature, NH₄OH (160 mL) was added dropwise in an ice water bath, chloroform (285 mL) was then added and the mixture was stirred overnight. The organic phase was separated and the aqueous phase was extracted with chloroform (3 × 100 mL). The organic phase was combined, filtered over Celite, and then concentrated to a volume of 50 mL. After ethanol (50 mL) and NH₄OH (10 mL) were added, the mixture was concentrated to a volume of 25 mL. The precipitated purple solid was filtered, washed with methanol (3 × 10 mL), and dried to give a solid containing **1** and its other three isomers. In a 250 mL three-necked flask, 36 g of silica gel (300–400 mesh) and benzene (85 mL) were added under N₂ atmosphere. After the mixture was stirred at 78 °C for 2 h, the solid containing **1** and its isomers was added and the mixture was further refluxed for 20 h. After cooling to room temperature, the dark slurry was then poured into a silica gel column. Benzene/diethyl ether = 1:1 was used to elute the three isomers of **1** and acetone/diethyl ether = 1: 1 was then used to elute **1** (650 mg, 32%). ¹H NMR (400 MHz, DMSO-*d₆*): δ 8.80 (s, 8 H), 7.68 (d, *J* = 7.3 Hz, 4 H), 7.52 (t, *J* = 7.7 Hz, 4 H), 7.15 (d, *J* = 8.2 Hz, 4 H), 7.00 (t, *J* = 7.3 Hz, 4 H), 4.61 (s, 8 H), and -2.72 (s, 2 H). HRESI-MS: *m/z* calcd for [M]⁺ C₄₄H₃₄N₈, 674.2906, found 674.2902, error 0.6 ppm.

**Synthesis of compound 4**. Triethylamine (14.0 mL, 100 mmol) was added to a solution of tetraethyleneglycol monomethyl ether (10.4 g, 50.0 mmol) in dichloromethane (80 mL) and the mixture was cooled to 0 °C. A solution of *p*-toluene-sulfonyl chloride (15.3 g, 80.0 mmol) in dichloromethane (50.0 mL) was added dropwise and then reacted for further 12 h at room temperature. After dichloromethane was removed under vacuum, the residue was purified by flash column chromatography with a mixture of petroleum ether and ethyl acetate as eluent (1: 1, v/v) to afford the compound **3** (12.5 g, 69%) as a colorless liquid. ¹H NMR (500 MHz, CDCl₃): δ 7.77 (d, *J* = 8.2 Hz, 2 H), 7.32 (d, *J* = 8.2 Hz, 2 H), 4.14 (t, *J* = 5.0 Hz, 2 H), 3.66 (t, *J* = 5.0 Hz, 2 H), 3.63–3.58 (m, 6 H), 3.56 (s, 4 H), 3.53–3.50 (m, 2 H), 3.35 (s, 3 H), and 2.42 (s, 3 H). ¹³C NMR (126 MHz, CDCl₃): δ 144.89, 133.15, 129.92, 128.06, 72.02, 70.67, 69.36, 68.76, 59.09, and 21.71.

The tosylate **3** (7.2 g, 20 mmol) was added to a mixture of 4-hydroxyacetanilide (3.6 g, 24 mmol), anhydrous potassium carbonate (4.1 g, 30 mmol), sodium iodide (50 mg), and acetonitrile (80 mL). The mixture was stirred at reflux for 12 h. After the reaction mixture was cooled to room temperature, it was filtrated and the solid was washed with dichloromethane (50 mL). The filtrate was concentrated under vacuum and the residue was dissolved in a KOH (11.2 g, 200 mmol) solution in ethanol (40 mL) and water (10 mL). After the mixture was stirred for 6 h at reflux, the solvents were removed under vacuum and the residue was purified by flash column chromatography (eluent: petroleum ether/EtOAc = 1: 3) to afford the compound **4** (3.4 g, 57% yield) as a colorless oil. ¹H NMR (500 MHz, CDCl₃): δ 6.75 (d, *J* = 8.8 Hz, 2 H), 6.62 (d, *J* = 8.8 Hz, 2 H), 4.04 (t, *J* = 5.0 Hz, 2 H), 3.80 (t, *J* = 5.0 Hz, 2 H), 3.74–3.68 (m, 2 H), 3.68–3.60 (m, 8 H), 3.54–3.52 (m, 3.7 Hz, 2 H), and 3.36 (s, 3 H). ¹³C NMR (126 MHz, CDCl₃): δ 151.75, 140.21, 116.15, 115.77, 71.82, 70.50, 69.78, 68.05, and 58.87.

**Synthesis of amphiphile 5**. To a solution of **1** (267 mg, 0.4 mmol) and Et₃N (0.6 mL) in dichloromethane (30 mL), a solution of triphosgene (179 mg, 0.6 mmol) in dichloromethane (30 mL) was added dropwise at 0 °C under nitrogen atmosphere. After the reaction solution was stirred for 3 h at room temperature, the intermediate **4** (1078 mg, 4.8 mmol) was added at 0 °C and the mixture was then stirred for further 2 h at room temperature. After water (30 mL) was added, the organic layer was separated, and the aqueous layer was further extracted with dichloromethane (20 mL). The combined organic solution was washed with brine (3 × 30 mL), and dried over anhydrous Na₂SO₄. The filtrate was concentrated under vacuum and the residue was purified by flash column chromatography (eluent:dichloromethane/methanol = 100:1) to afford the amphiphilic porphyrin **5** (539 mg, 68% yield) as a purple solid. M.p. 108–110 °C. ¹H NMR (500 MHz, DMSO-*d₆*) δ 8.78 (s, 8H), 8.41 (d, *J* = 8.0 Hz, 4 H), 8.32 (s, 4 H), 7.77 (t, *J* = 7.7 Hz, 4 H), 7.68 (d, *J* = 6.8 Hz, 4 H), 7.59 (s, 4 H), 7.37 (t, *J* = 7.3 Hz, 4 H), 6.84 (d, *J* = 8.5 Hz, 8 H), 6.51 (d, *J* = 8.5 Hz, 8 H), 3.80 (s, 8 H), 3.57 (s, 8 H), 3.50–3.34

(m, 48 H), 3.16 (s, 12 H), and −2.64 (s, 2 H). $^{13}$C NMR (126 MHz, DMSO-$d_6$) $\delta$ 153.81, 153.14, 150.58, 139.60, 135.89, 132.77, 131.84, 129.57, 122.19, 121.88, 119.85, 116.23, 114.80, 99.99, 71.68, 70.30, 70.21, 70.18, 69.99, 69.34, 67.53, 58.44, and 46.24. HRESI-MS: $m/z$ Calcd for $C_{108}H_{126}N_{12}O_{12}$ [M + H]$^+$: 1975.9008, Found:1975.9080, error 3.6 ppm. IR (thin film): $\nu_{max}$ (cm$^{-1}$) = 3550, 3474, 3414, 3126, 1635, 1616, and 1400.

**Synthesis of amphiphile 6**. To a solution of **5** (40 mg, 0.020 mmol) in CHCl$_3$ (30 mL) was added Zn(OAc)$_2$ (200 mg, 1.099 mmol), and the mixture was stirred for 3 h at 60 °C. After the mixture was evaporated to remove solvents, the residue was purified by flash column chromatography (eluent:dichloromethane/methanol=100:1) to afford the amphiphilic Zn-porphyrin **6** (30 mg, 73% yield) as a purple solid. M.p. 112–115 °C. $^1$H NMR (500 MHz, DMSO-$d_6$) $\delta$ 8.70 (s, 8 H), 8.42 (s, 4 H), 7.83–7.72 (m, 8 H), 7.38 (s, 4 H), 6.67 (s, 8 H), 6.32 (s, 8 H), 3.72 (s, 8 H), 3.55 (s, 8 H), 3.44–3.32 (m, 48 H), and 3.16 (s, 12 H). $^{13}$C NMR (126 MHz, DMSO-$d_6$) $\delta$ 153.12, 152.47, 149.63, 139.20, 134.81, 132.29, 132.19, 131.52, 128.50, 120.83, 119.45, 119.35, 115.36, 114.12, 71.17, 69.78, 69.70, 69.66, 69.47, 68.80, 67.00, and 57.92. HRESI-MS: $m/z$ Calcd for $C_{108}H_{124}N_{12}O_{12}Zn$ [M + H]$^+$: 2037.8143, found: 2037.8174, error 1.5 ppm. IR (thin film): $\nu_{max}$ (cm$^{-1}$) = 3565, 3550, 3473, 3416, 3126, 1652, 1616, and 1400.

**Preparation of TEM, SEM, and AFM samples**. Amphiphilic porphyrin **5** or its Zn-porphyrin derivative **6** and the corresponding aromatic compounds (**7** or **8**) or pyridine templates (**9** or **10**) were dissolved in THF or in a mixed organic solvent and the solutions were mixed in a specific mole ratio. The mixtures were then added into the deionized water and stirred overnight until THF or other organic solvents were volatilized completely. Drops of each solution were cast on carbon-coated grids (Cu, 400 mesh) and dried naturally for two days before TEM photography. Drops of these mixed solutions were cast on monocrystalline silicon, dried naturally for 2 days and sprayed gold before SEM photography, or cast on monocrystalline silicon and dried naturally for 2 days before AFM photography. More detailed procedures of sample preparation for electron microscope are described in Supplementary Methods.

## Data availability
The authors declare that the data supporting the findings of this study are available within the paper and its Supplementary Information file.

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

## Acknowledgements
This work was supported by the National Natural Science Foundation of China (21572042 and 21773052), the Program for Innovative Research Team in Chinese University (IRT 1231), the Natural Science Foundation of Zhejiang Province (LZ16B020002 and LQ17B040002), and the Program for Social Development of Hangzhou (20170533B10).

## Author contributions
S.L. and J.C.H. conceived and designed the experiments. S.-P.W. and W.L. performed the synthesis and self-assembly experiments. S.-P.W., X.W., T.-Y.C., and B.-Y.Z. conducted the complexation investigation, characterization of supra-amphiphiles, and electron microscope studies. H.X. and J.Y.H. carried out the theoretical calculations. Z.Z., A.S., J.C.H., J.W., and S.L. analyzed the data and wrote the manuscript together.

## Additional information

**Competing interests:** The authors declare no competing interests.

