## [Peer Review File · Nature Communications]

Reviewers' comments:

Reviewer #1 (Remarks to the Author):

In this paper, the authors fabricated a series of supra-amphiphiles based on amphiphilic porphyrin derivatives. By changing the guest molecules, including Cl⁻, coronene, C60, 4,4'-bipyridine and 2,4,6-tri(pyridin-4-yl)-1,3,5-triazine, the formed supra-amphiphiles self-assembled into various nano/micro structures. The work was studied in detail. However, numerous works about the self-assembly of supra-amphiphiles with different building blocks, shape, topology, and driving forces have already been studied. The authors should tell the readers what's new and what can we learn from this work. In addition, the readers may wonder if these nano/micro structures could lead to some fancy properties and functions. Considering the novelty and completeness of this work, I cannot recommend it for publication in Nature communications in the present form. Some questions are listed below:

1. In general, the binding affinity between building blocks is the key factor which influences the stability of formed supra-amphiphile. However, there is little quantitative description about the binding affinity of the amphiphilic porphyrin (5 and 6) toward the co-assembly guest molecules (Cl⁻, 7, 8, 9, 10) in this manuscript, the relevant quantitative experiments should be supplemented.

2. When discussing the reason why addition of Cl⁻ can promote the formation of well-arranged supramolecular structures, authors said: "This can be attributed to the strong affinity of ureas on the amphiphilic porphyrin toward the spherical Cl⁻ anions via host-guest complexation that increases intermolecular interactions of 5 to promote the formation of larger self-assemblies". The ¹H NMR experiment was employed to confirm their complexation.

-Firstly, the statement "host-guest complexation" is suggested to be reconsidered.

-Secondly, the results of ¹H NMR experiments present the change of chemical shifts. However, what is the logical relationship between this result and the proposed judgement?

-Thirdly, how to eliminate the interference of the counterion of n-Bu₄N⁺? How about addition of HCl or NaCl? The control experiment should be provided.

3. It is noted that the solutions used in supramolecular assembly process are different from that used in ¹H NMR experiments. It is suggested that the solution should be kept the same in one system.

Reviewer #2 (Remarks to the Author):

The manuscript by Li and coworkers presents the synthesis of amphiphilic porphyrins grafted with oligoether chains, and their self assembly behavior upon the addition of aromatic guest molecules. It is very interesting that different morphologies including ribbons, membranes, spheres, and long fibers can be made by using different guest molecules. The synthesis, the structural characterization, and the analysis of the assembly mechanisms are well described in the manuscript and convincing. To this reviewer, there are two major achievements in this work: first, the clever use of guest molecules to regulate the self-assembly of amphiphilic molecules; second, the organization of porphyrins into well defined nanostructures with good local orderings. Porphyrins are a class of functional molecules that are particularly interesting due to their photochemical and photophysical properties. Developing a simple approach to make composite nanomaterials that can incorporate other functional molecules such as coronene, C60 into porphyrin-based assemblies is highly desired. The technical quality of the manuscript is excellent. I recommend the publication of this manuscript in Nature Communications after minor revision. I have a few comments and suggestions, which I would like the authors to consider upon revising their manuscript:

1. For supramolecular polymers assembled from synthetic subunits in solution, the coexistence of multiple polymorphs is often unavoidable. This also seems to be the case, judged from the TEM images shown in the manuscript. As uncontrolled polymorphism will be an obstacle for future

progress of controlled synthesis of supramolecular polymers and materials, all parameters of the pathway control must be optimized to obtain well-defined structures with a good yield. The authors mainly rely on the change of solvent conditions to prepare the assembled structures (e.g., from organic solvent to a mixture of organic solvent and water, and eventually let the organic solvent volatilized). How to optimize this process for well-defined structures, and how do we know the structures are thermodynamically stable structure instead of kinetically trapped structures due to the preparation procedure?

2. In addition, among different experimental parameters, optimizing temperature is particularly challenging as it requires the characterization and analysis of the self-assembled aggregate across a broad range of temperatures and times. In some cases, even the cooling/heating rate can bias one assembly pathway over the other. For the benefit of the readers, can the authors comment on the temperature effect on their system and add some discussion in the manuscript?

3. It is not clear to this reviewer whether the structures in Figure 1 are rods or tapes. This should be relatively easy to characterize by zooming in the FESEM images at the higher magnification or by AFM. In general, Figure 1 and 7 are too small. It is difficult to see the structural details in some of the subfigures.

4. Does the ordering of porphyrins vary in the different morphologies? Electron diffraction patterns may be collected from the TEM samples and be compared to answer this question.

5. The references can be more selective, in order to help the readers focus on the most relevant works in the field.

Reviewer #3 (Remarks to the Author):

The manuscript by Wang et al. is an interesting piece of work. The novelty is in the control of shape, dimension and size of supramolecular architectures upon introduction of different species (i.e., chloride anions, fullerene, 4,4'-bipyridine, triazine derivative). Although specific interpretation of the reasons why a specific supramolecular assembly is preferred depending on the various species added is missing, the results are anyway clear and very interesting, in the light of the potential developments. Indeed, experimental results of the presented kind are more and more needed to help to rationalize the behavior, that is still hard to understand nowadays.

In the light of the potential developments and considering the detailed important results, I am in favor of publication of the present paper in Nature Communications. The paper is also prepared with care, I did not found points to be corrected.

We appreciate the reviewers' comments which have greatly improved our manuscript. The specific changes to the manuscript as per request are listed as following.

Reviewer 1:

“However, numerous works about the self-assembly of supra-amphiphiles with different building blocks, shape, topology, and driving forces have already been studied. The authors should tell the readers what’s new and what can we learn from this work. In addition, the readers may wonder if these nano/micro structures could lead to some fancy properties and functions.”

As the reviewer mentioned, there are many literatures on supra-amphiphiles, as seen in the references 4–32 and the references in those publications. Nevertheless, porphyrin-based recognition motifs have seldom been utilized to construct supra-amphiphiles and rare of them provided so much abundant control on the shapes and dimensions of the self-assemblies. An emphasis on the importance of this manuscript, “which highlights the present work with the abundant control on the shapes and dimensions of the self-assemblies” was correspondingly added in the end of Abstract.

Since these nano/micro structures consist of porphyrins, C₆₀ or other aromatic molecules with good photophysical and electrochemical properties, they may have potential applications in photoelectrical materials. In especial, the lamellar self-assembly that possessing thin layers of C₆₀ should be very attractive for the construction of photovoltaic materials. A relative description, “This strategy provides a highly efficient way to fabricate monolayers of fullerenes, which is quite attractive for the application in photovoltaic materials”, has been added in the end of the paragraph in Page 8.

“In general, the binding affinity between building blocks is the key factor which influences the stability of formed supra-amphiphile. However, there is little quantitative description about the binding affinity of the amphiphilic porphyrin (5 and 6) toward the co-assembly guest molecules (Cl⁻, 7, 8, 9, 10) in this manuscript, the relevant quantitative experiments should be supplemented.”

Determination of association constants between the amphiphilic porphyrin (**5** and **6**) and the guests (**7**, **8**, **9** and **10**) has been supplemented. The Benesi-Hildebrand plots and Scatchard plots for determination of association constants were added on Page S22–S25, in the Supplementary Information. The complexation between the amphiphilic porphyrin (**5** and **6**) and Cl⁻ was found too complicated to determine the association constants. As the data shown, the complexation sites behave independently in all of these cases. K_1 and K_2 of **5**₂•**7** in THF were measured to be $8.5 (\pm 0.2) \times 10^4 \text{ M}^{-1}$ and $2.1 (\pm 0.1) \times 10^4 \text{ M}^{-1}$, respectively. K_1 and K_2 of **5**₂•**8** in THF were determined to be $3.2 (\pm 0.2) \times 10^3 \text{ M}^{-1}$ and $8.0 (\pm 0.4) \times 10^2 \text{ M}^{-1}$, respectively. K_1 and K_2 of **6**₂•**9** in THF were measured to be $1.6 (\pm 0.1) \times 10^3 \text{ M}^{-1}$ and $4.0 (\pm 0.2) \times$

10^2 M^{-1} , respectively. K_1 , K_2 and K_3 of **63•10** in THF are $2.4 (\pm 0.1) \times 10^3 \text{ M}^{-1}$, $8.1 (\pm 0.1) \times 10^2 \text{ M}^{-1}$ and $2.7 (\pm 0.1) \times 10^2 \text{ M}^{-1}$, respectively. These considerable binding affinities are conducive to the formation of stable supra-amphiphiles. The data and the corresponding discussion have been added in the main text and Supplementary Information.

“When discussing the reason why addition of Cl^- can promote the formation of well-arranged supramolecular structures, authors said: “This can be attributed to the strong affinity of ureas on the amphiphilic porphyrin toward the spherical Cl^- anions via host-guest complexation that increases intermolecular interactions of 5 to promote the formation of larger self-assemblies”. The ^1H NMR experiment was employed to confirm their complexation.

-Firstly, the statement “host-guest complexation” is suggested to be reconsidered.

-Secondly, the results of ^1H NMR experiments present the change of chemical shifts. However, what is the logical relationship between this result and the proposed judgement?

-Thirdly, how to eliminate the interference of the counterion of $n\text{-Bu}_4\text{N}^+$? How about addition of HCl or NaCl ? The control experiment should be provided.”

The statement “host-guest complexation” and similar description have been changed to “recognition motif” or “complexation” through all of the main text and Supplementary Information.

As to the second point, the change of chemical shifts, as well as the change of other physical properties, such as UV-vis absorption and fluorescence, in a mixture of non-reactive species generally means the existence of complexation or supramolecular interactions, which has been widely used to judge the occurrence of complexation. The existence of NOE signals in NOESY spectra provided further evidence for the complexation between the amphiphilic porphyrin and the co-assembly guests in this manuscript. As seen in the above reply, the quantitative studies on determination of association constants in THF have also been supplemented to support their complexation.

Thirdly, the counterion of $n\text{-Bu}_4\text{N}^+$ was used because of its good solubility in organic solvents and its loose ion pair characteristics, which can benefit the complexation of Cl^- with the urea groups on **5** or **6**. This quaternary ammonium counterion was extensively used in the recognition of anions owing to these advantages (seeing the reference 58). NaCl , as a highly intimate ion pair, is much more difficult to be complexed by the hosts than that the loose ion pairs do. We have supplemented the control experiments, proving that NaCl cannot be recognized by the amphiphilic molecules, as seen in Figure S10 and S11. HCl is usually used as an acid and simultaneously introducing the chloride anion, but seldom merely used as a source of chloride anion, so it was not used.

“It is noted that the solutions used in supramolecular assembly process are different from that used in ¹H NMR experiments. It is suggested that the solution should be kept the same in one system.”

We have supplemented NMR studies in THF-*d*₈ solutions, from which similar results were obtained as they had been done in CDCl₃. The ¹H NMR data and the corresponding discuss have been changed in the main text and Supplementary Information. Now the solvents in the ¹H NMR comparison experiments are same as those for the supramolecular assembly, except for in the system of **5**₂•**8** due to the low solubility of C₆₀ in THF-*d*₈.

Reviewer 2

“For supramolecular polymers assembled from synthetic subunits in solution, the coexistence of multiple polymorphs is often unavoidable. This also seems to be the case, judged from the TEM images shown in the manuscript. As uncontrolled polymorphism will be an obstacle for future progress of controlled synthesis of supramolecular polymers and materials, all parameters of the pathway control must be optimized to obtain well-defined structures with a good yield. The authors mainly rely on the change of solvent conditions to prepare the assembled structures (e.g., from organic solvent to a mixture of organic solvent and water, and eventually let the organic solvent volatized). How to optimize this process for well-defined structures, and how do we know the structures are thermodynamically stable structure instead of kinetically trapped structures due to the preparation procedure?”

As the reviewer pointed out, optimization of experimental parameters during self-assembly is important to obtain well-defined structures. We have done the optimization of parameters, including solvent, temperature, and time. It was found that THF was the best choice for self-assembly among the screened organic solvents, including acetone, acetonitrile, chloroform, dichloromethane and ethanol, probably attributing to the good solubility of amphiphilic porphyrin in THF and good miscibility of THF with water. A corresponding description has been added in the beginning of the last paragraph on Page 5.

As the self-assemblies over longer time and at higher temperature revealed, the obtained aggregates of the samples prepared after 3 days and 7 days at room temperature, as well as the samples prepared at 60 °C showed no significant difference in aggregation morphologies, which implies the observed aggregates are thermodynamically stable structures, but not kinetically trapped structures. The results and a corresponding description have been added in the first paragraph on Page 12 in the main text and Page S30–S32 in Supplementary Information.

“In addition, among different experimental parameters, optimizing temperature is particularly challenging as it requires the characterization and analysis of the self-assembled aggregate across a broad range of temperatures and times. In some cases, even the cooling/heating rate can bias one assembly pathway over the other.

For the benefit of the readers, can the authors comment on the temperature effect on their system and add some discussion in the manuscript?"

The assembly process of our system was found not sensitive to temperature. The control experiments demonstrated that the shape of the assembly at room temperature and at 60 °C did not change obviously. The results and a corresponding description have been added in the first paragraph on Page 12 in the main text and Page S30–S32 in Supplementary Information.

"It is not clear to this reviewer whether the structures in Figure 1 are rods or tapes. This should be relatively easy to characterize by zooming in the FESEM images at the higher magnification or by AFM. In general, Figure 1 and 7 are too small. It is difficult to see the structural details in some of the subfigures."

We have done the scanning electron microscope (SEM) again and the images revealed the structures made from **5₂•7** are rectangular rods. The images have been replaced.

"Does the ordering of porphyrins vary in the different morphologies? Electron diffraction patterns may be collected from the TEM samples and be compared to answer this question."

High-resolution TEM (HRTEM) and selected area electron diffraction (SAED) experiments have been supplemented to understand the internal structures of these aggregates. As revealed by the HRTEM image and strong SAED diffraction patterns (Fig. 8a and 8b), the nanorods self-assembled from **5₂•7** show good crystallinity of the micro-sized crystals. The membranes constructed from **5₂•8** have weak crystallinity (Fig. 8c and 8d) while the micelles self-assembled from **6₂•9** have no diffraction and thus are not crystalline (Fig. 8e and 8f). The consecutive diffraction patterns from the microwires that aggregated by **6₃•10** implies the existence of superstructures or subunits that can also be directly observed in the enlarged HRTEM picture (Fig. 8g and 8h), the results of which provides further solid evidence for the self-assembly mechanism deduced from the TEM and SEM observations at different stages. Overall, the varied diffraction patterns from the aggregates with different morphologies suggest their different arrangement modes of building molecules. The corresponding description has been added in the main text on Page 14–15.

"The references can be more selective, in order to help the readers focus on the most relevant works in the field."

We have deleted the references that are not highly relevant to this work, ref. 14, 15, 17, 43, 45, and 46 marked in the previous manuscript. Meanwhile, a couple of new references, that are necessary to understand the determination of binding constants and SAED, were added.

REVIEWERS' COMMENTS:

Reviewer #1 (Remarks to the Author):

The authors have made significant improvements. The manuscript may be acceptable after the CACs of the supra-amphiphiles are determined. Since the binding constants of the complexation between building blocks are on the order of 10^3 to 10^4 M⁻¹, the stability of the supra-amphiphiles at low concentrations should be discussed.

Reviewer #2 (Remarks to the Author):

The revision of authors and the clarifications in the rebuttal letter fully addressed this reviewer's concern. I would like recommend the publication of the manuscript in Nature Communications.

We appreciate the reviewers' comments which have greatly improved our manuscript. The specific changes to the manuscript as per request are listed as following.

Reviewer 1:

“The authors have made significant improvements. The manuscript may be acceptable after the CACs of the supra-amphiphiles are determined. Since the binding constants of the complexation between building blocks are on the order of 10^3 to 10^4 M⁻¹, the stability of the supra-amphiphiles at low concentrations should be discussed.”

Thanks for your comments. The CACs of the supra-amphiphiles **5₂•7**, **5₂•8**, **6₂•9** and **6₃•10** have determined to be 5.5×10^{-6} M, 8.0×10^{-6} M, 1.8×10^{-5} M, and 1.7×10^{-5} M, respectively. These values are close each other and close to the CAC values of the amphiphiles **5** and **6** that had previously measured and provided in the manuscript. These results and corresponding discuss have been added in the main text, as follows: “Based on the UV-vis transmittance, the CACs of supra-amphiphiles **5₂•7**, **5₂•8**, **6₂•9** and **6₃•10** in aqueous solutions were measured to be 5.5×10^{-6} M, 8.0×10^{-6} M, 1.8×10^{-5} M, and 1.7×10^{-5} M, respectively (Supplementary Fig. 25–28). These supra-amphiphiles with different geometry have rather close CAC values, which indicates that structural characteristics and supramolecular interactions, but not the difference in hydrophilicity or lipophilicity, play important roles on the formation of varied aggregation morphologies.”

A discussion on the stability of the supra-amphiphiles at low concentrations has been added in the main text, as following: “In spite of the fact that uncomplexed species usually increase at low concentrations, hydrophobic interactions existed during self-assembly in aqueous solutions would greatly enhance the stability of the supra-amphiphiles.”